# Benchmarking spike source localization algorithms in high density probes

**Hao Zhao[1]\*, Xinhe Zhang[1,2]\*, Arnau Marin-Llobet[1], Xinyi Lin[1], Jia Liu[1]**

**1** John A. Paulson School of Engineering and Applied Sciences, Harvard University, Boston, Massachusetts, United States of America, **2** Broad Institute of MIT and Harvard, Cambridge, Massachusetts, United States of America

\* hzhao@mba2024.hbs.edu (HZ), xinhezhang@g.harvard.edu (XZ)

## Abstract

Estimating neuron location from extracellular recordings is essential for developing advanced brain-machine interfaces. Accurate neuron localization improves spike sorting, which involves detecting action potentials and assigning them to individual neurons. It also assists in monitoring probe drift, which affects long-term probe reliability. Although several localization algorithms are currently in use, the field is nascent and arguments for using one algorithm over another are largely theoretical or based on visual inspection of clustering results. We present a first-of-its-kind benchmarking of commonly used neuron localization algorithms. We assess these algorithms using two ground truth datasets: a biophysically realistic simulated dataset, and an experimental dataset pairing patch-clamp and extracellular Neuropixels recording data. We systematically evaluate the accuracy, robustness, and runtime of these algorithms in ideal recording conditions and long-term recording conditions with electrode degradation. Our findings highlight significant performance differences; while more complex and physically realistic models perform better in ideal conditions, models relying on simpler heuristics demonstrate superior robustness to noise and electrode degradation, making them more suitable for long-term neural recordings. This work provides a framework for assessing localization algorithms and developing robust, biologically grounded algorithms to advance the development of brain-machine interfaces.

## Author summary

Accurately estimating neuron locations from extracellular recordings is critical to building reliable brain–machine interfaces. This spatial information enhances spike sorting and enables long-term monitoring of neural activity, especially in the presence of probe drift and electrode degradation. Despite the availability of several spike source localization algorithms, their comparative long-term performance has not been systematically benchmarked against ground truth data. In

**Data availability statement:** All analysis scripts and supporting functions used in this study are available on our github repository (https://github.com/haozhao1996/Spike-Localization-Algorithms), where we have additionally included the intermediate data including spike trains, waveforms, and localization values for both the simulated and experimental datasets (https://doi.org/10.6084/m9.figshare.31566010). The code is released under the MIT License, permitting reuse, modification, and distribution with attribution. The source code for generating the simulated dataset (https://github.com/SpikeInterface/MEArec), and the experimental dataset (https://crcns.org/data-sets/methods/spe-1/about-spe-1), have been created and are maintained by other institutions.

**Funding:** A.ML. acknowledges support from the RCC Fellowship of Harvard University and the Excellence Fellowship of the Fundacion Rafael del Pino. The funders had no role in study design, data collection and analysis, decision to publish, or preparation of the manuscript.".

**Competing interests:** I have read the journal's policy and the authors of this manuscript have the following competing interests: J.L. is cofounder of Axoft, Elastro, AIScientists, and NanoRhythmics.

this study, we benchmark three widely used algorithms—center of mass (COM), monopolar triangulation (MT), and grid convolution (GC)—using both simulated and experimental ground truth datasets. We assess their accuracy, runtime, and robustness under ideal and degraded recording conditions. Our results reveal that while MT demonstrates higher accuracy in ideal conditions, GC and COM demonstrate superior resilience to noise and electrode degradation, making them more suitable than MT for long-term recordings. These findings provide a foundational framework for evaluating and improving spike localization algorithms and highlight the importance of robustness in real-world neural interface applications.

## Introduction

Improvements in high-density multi-electrode arrays (MEAs) in the past several years have enabled the extracellular study of neuron populations over long periods of time [1,2]. A key advantage of high-density MEAs is the spatial density of electrodes, which allows for the same action potential (or "spike") to be detected at multiple electrode sites, providing an opportunity to estimate the location of each spike. This spatial information is important in a variety of downstream tasks. One of these key tasks is "spike sorting", which is the process of detecting action potentials and assigning each spike to an individual neuron or "unit". Accurate spike sorting is crucial for understanding neural circuits, as incorrectly assigned spikes can significantly distort the interpretation of neural activity [3]. Currently, state-of-the-art spike sorting algorithms and strategies rely heavily on spatial information to enhance their accuracy [4–8]. Following the spike sorting process, the estimated location of the neuron (based on the aggregate signal of all spikes assigned to that neuron, referred to as a "template") is useful for the evaluation of brain-machine interface performance (e.g., to assess probe drift) [9].

While several localization algorithms are currently used, there has been no ground-truth benchmarking of their performance and robustness against degradation (i.e., loss of recording signal from electrode sites on a probe over time) [9–11]. The most widely used localization algorithm is center of mass ("COM"), which uses a "weighted average" heuristic that is simple but physically inaccurate. Other more physically accurate algorithms have been proposed in recent years, including monopolar triangulation ("MT") and grid convolution ("GC"); along with COM, these methods are implemented in the widely used SpikeInterface package [12]. However, despite significant variance in the algorithms and their underlying physical assumptions, there has been no systematic analysis to date of their performance in long-term recording situations.

A better understanding of performance is particularly important given advancements in neural probe technology. Current probes, such as those which are designed with tissue-level flexibility, are able to record from the same neuron populations over periods of up to over a year [13–19,20]. Over long-term recording horizons, electrodes within

the array will degrade, and localization algorithms should be robust against this loss of signal if they are to be suitable for decoding long-term brain recordings and accurately tracking the same neurons over long periods of time. Additionally, probes with tissue-level flexibility reduce the mechanical and structural differences between the probe and brain tissue, and appropriate localization algorithms are important to assess the degree of improvement in probe drift [2,19,21,22].

To address this lack of understanding, we perform a first-of-its-kind benchmarking of COM, MT, and GC against two sources of ground truth data and electrode degradation. The first is a biophysically realistic simulated dataset, which has recently been used to benchmark stabilization algorithms for probe drift (a separate but related problem) [5,23]. The second is an experimental dataset of paired recordings where the same neurons were recorded from patch clamp (providing the "ground truth" spike train) and extracellular MEA (Neuropixel 384-channel probe), and the approximate neuron location is known for each paired recording [24]. For each dataset, we first benchmark the performance of all three localization algorithms on both spikes and templates in the absence of electrode degradation. Then, we simulate electrode degradation and examine how robust the different localization algorithms are to signal loss.

## Data and methods

### Simulated dataset

The simulated dataset uses the MEArec simulator to generate ground truth recordings using biophysically detailed multi-compartment models, and follows similar protocols to previous analyses used to simulate ground truth conditions [5,23]. We use a dictionary of 13 cell models (representing a mix of excitatory and inhibitory neurons) from layer 5 of a juvenile rat somatosensory cortex, provided via the Neocortical Microcircuit Collaboration portal [25,26]. We use these cell models to simulate a dictionary of biologically plausible templates on a Neuropixel 384-channel probe. For each recording, we simulate 50 neurons selected from our template dictionary and generate corresponding spike trains. Templates and spike trains are then convolved, adding a slight modulation in amplitude to add physiological variability. Uncorrelated Gaussian noise with 10 µV standard deviation is then added to the traces. The sampling rate of the simulated recordings is 30 kHz. From the simulated dataset, we know the ground-truth neuron locations and spike trains used to generate the recording data, and we use these ground-truth spike trains to extract spike waveforms and template waveforms (constructed using the median across spike waveforms for a given neuron) at each recording electrode. The simulation length for each level of degradation was 90 seconds, and 25 levels of degradation (from 0 percent to 94 percent) were explored (i.e., 2,250 seconds of recordings across "life" of degraded probe in each trial). We ran simulations initialized to five different starting arrangements for the 50 simulated neurons, and with five different degradation patterns (i.e., 25 trials).

### Experimental dataset

For experimental benchmarking, we use the open-source SPE-1 dataset of paired patch clamp and MEA (Neuropixel 384-channel) recordings from the same cortical neurons of anaesthetized rats [24]. The patch clamp recordings are used to determine the ground-truth spike train for the neuron, which we use to extract spike waveforms and template waveforms at each recording electrode. In this dataset, the authors additionally tracked the relative location of the patch-clamp to the MEA in each paired recording, and were able to approximate the closest electrode on the MEA to the patch clamp, providing an estimate of ground-truth neuron location. The dataset consists of 43 paired recordings, of which we select 11 pairs with the strongest signal quality and confidence of patch-clamp and MEA pairing.

### Electrode degradation

In order to simulate electrode degradation, we replace a progressively increasing proportion of electrodes with uncorrelated 50µV Gaussian noise, intended to represent channels that lose functionality over a long-term period (e.g., physical damage or disconnection of site, material gliosis causing high impedance around site, etc.).

## Pre-processing

A spike sorting pre-processing pipeline for denoising and anomalous channel detection was applied to both the simulated and experimental datasets. A 300–3,000 Hz Butterworth bandpass filter (fifth-order, causal filter) was applied to the raw recordings, followed by common median referencing. We then perform anomalous channel detection using the established IBL protocol, detecting anomalous channels via coherence (i.e., adjacent channels within brain tissue should have a high degree of signal correlation) and power distribution (i.e., healthy channels should have a normal power distribution across frequencies) [27].

## Spike and template extraction

Following pre-processing of the extracellular recordings, we use the ground-truth spike trains to extract spike waveforms without relying on a spike sorting algorithm. Leveraging the known ground-truth spike trains allows us to avoid bias which would be introduced by spike sorting (which generally relies on some assumption about spike localization) and reduces noise (e.g., detection thresholds, clustering errors, or missed events), and so isolates our analysis to the basic spike localization problem. For each neuron, we collect short segments of the preprocessed signal centered on each ground-truth spike, using a fixed temporal window so that the full spike waveform and its immediate context are captured. Waveforms are then aligned to the peak, to reduce temporal jitter and to isolate the characteristic shape of each neuron's extracellular signature. We then extract template waveforms by aggregating all spikes from the same neuron, in order to obtain a representative waveform that captures the temporal and spatial footprint of the neuron in a given trial.

## Localization algorithms

For spike $i$, we denote the spike waveform at each electrode $j$ as $w_{ij}(t)$. We define the peak-to-peak amplitude of $w_{ij}(t)$ as the voltage difference between the peak and the trough of the waveform, or $\text{ptp}_{ij}$. For brevity, we denote the spike location as $p_i = (x_i, y_i, z_i)$ and electrode location as $p_j = (x_j, y_j, z_j)$ (the subscript will indicate whether we are referring to a spike or electrode location). We will define $x$ and $y$ to lie along the two-dimensional plane of the MEA, and $z$ to lie on the vector orthogonal to the MEA. See Fig 1 for a visualization of these three localization algorithms.

**Center of Mass ("COM")**: The location of spike $i$ is estimated as:

$$p_i = (x_i, y_i, z_i) = \frac{\sum_j \text{ptp}_{ij} \cdot p_j}{\sum_j \text{ptp}_{ij}}$$

We note that COM has known theoretical limitations to estimating neuron location correctly, but it continues to be the most commonly used heuristic for spike source localization due to speed and simplicity.

**Monopolar Triangulation ("MT")**: Assumes that the neuron is a monopolar point source and voltage signal decays as an inverse of the distance from the neuron [9,28,29]. For spike $i$, the peak-to-peak voltage measured at each electrode can therefore be expressed as:

$$V_{ij} \simeq \frac{c_i}{\|p_i - p_j\|_2}$$

where $c_i$ is a unique constant and $\| \cdot \|_2$ is the Euclidean norm. For a multi-electrode array of $N$ electrodes, spike $i$ will have $N$ equations, representing $V_{ij}$ at $N$ electrodes. We define a loss function based on the difference between the estimated $V_{ij}$ and actual $\text{ptp}_{ij}$:

$$L_i = \sum_j (\text{ptp}_{ij} - V_{ij})$$

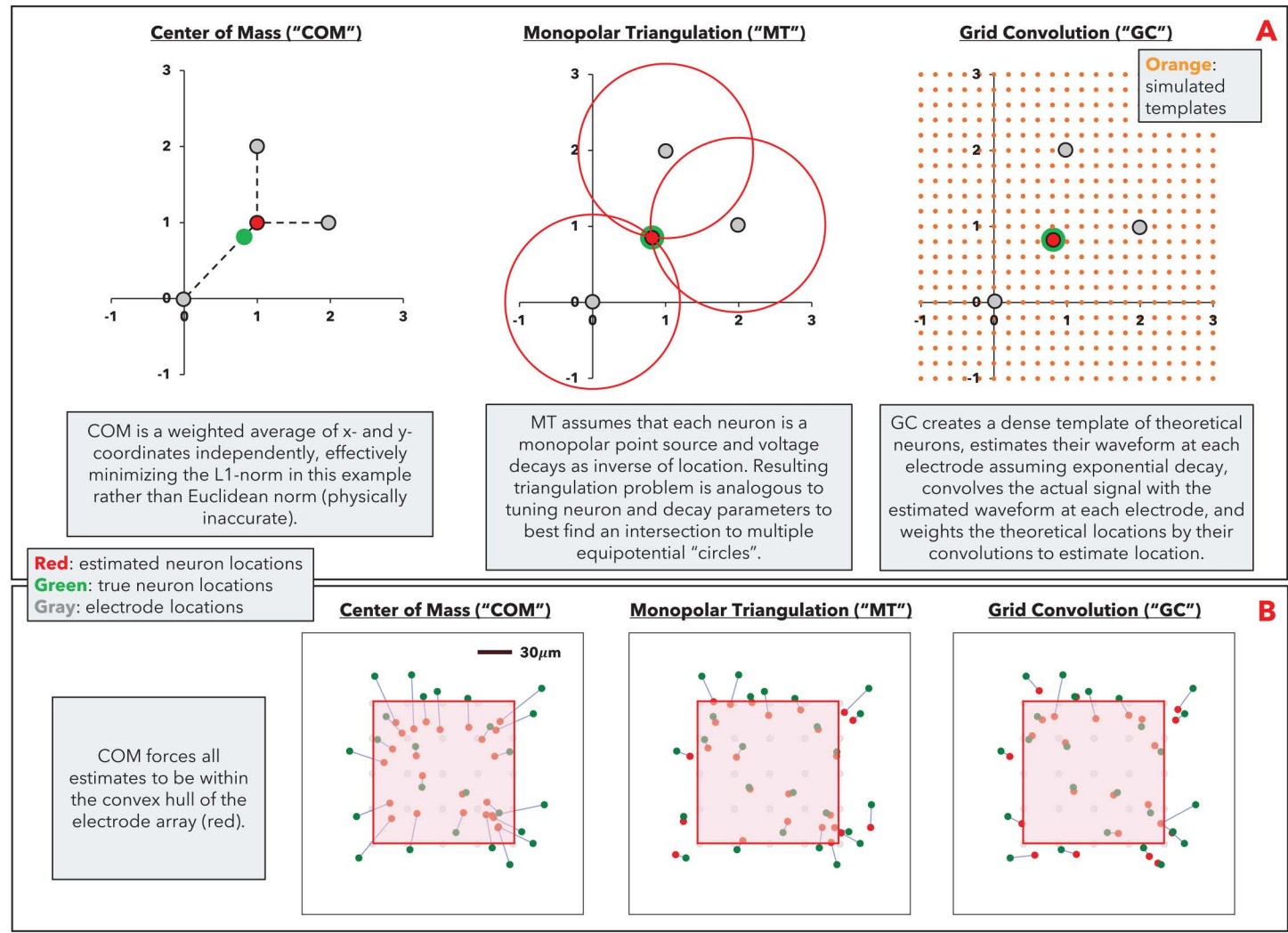

**Fig 1. Limitations of center of mass localization. A**: Visualization of localization algorithms for a point-source neuron equidistant to three electrodes located on a grid at (0,0), (1,2), and (2,1), where the electrodes will all receive the same signal strength from the neuron in ideal recording conditions (no noise or electrode degradation), given the neuron is located equidistant to all three electrodes. MT and GC are able to correctly estimate the "true" location while COM is not. **B**: Simulated electrophysiological data (MEArec), where green dots denote true neuron locations, red dots denote estimated neuron locations, and blue lines connect the true and estimated locations for a given neuron. Since COM is a weighted average of electrode locations, it forces all location estimates to be within the convex hull of the electrode array (all estimated neuron locations are within red square).

We then estimate the location of spike *i* as that which minimizes the loss function, using an optimization method of choice (e.g., least squares). We use the SpikeInterface implementation of this method, which uses a least-squares optimizer and several regularization parameters including limiting contributing channels to a local radius around the peak channel and limiting the location estimate to a candidate window around the center of mass estimate. We utilize SpikeInterface's default parameters (e.g., local radius of 75μm, candidate window of 1,000μm; please refer to source documentation for full parameter detail). We note that removing regularization negatively impacted MT performance; while removing regularization would better capture the mathematical properties of MT, we believe that the inclusion of regularization better reflects a real-world, practical implementation of the algorithm.

**Grid Convolution ("GC")**: Creates a dense grid of $k$ theoretical templates of waveform $w_k(t)$ located at positions $p_k$ (the grid is usually denser than the electrodes). For each of these theoretical templates, we simulate the signal that the template would produce on each electrode of the multi-electrode array, assuming an exponential decay:

$$\tau_{kj}(t) = e^{-(p_k - p_j)^2/(\sigma^{2.5})} w_k(t)$$

The dot product between $w_{ij}(t)$ and $\tau_{kj}(t)$ measures similarity; we can estimate the location of spike $i$ as a weighted sum of the theoretical template locations using this similarity measure [4]. We use the SpikeInterface implementation of this method, which includes regularization parameters including limiting the grid to a local radius around the peak channel and limiting the weighted sum to only consider a certain percentage of the strongest convolutions. We utilize SpikeInterface's default parameters (e.g., local radius of 40μm, weighted sum considering top 5% of convolutions, up-sampled grid resolution of 5μm; please refer to source documentation for full parameter detail). We note that removing regularization negatively impacted GC performance, though is not as impactful as it is to MT; while this would better capture the mathematical properties of GC, we believe that the inclusion of regularization better reflects a real-world, practical implementation of the algorithm.

## Results

### Benchmarking results in absence of electrode degradation

We benchmarked all three localization algorithms using the simulated ground truth dataset and experimental ground truth dataset without electrode degradation. In the visualization of these localization results, we see hallmarks of each algorithm consistent with prior visual analyses (Fig 2A and 2B). We see that COM is only able to produce location estimates within the convex hull of the electrode array, and tends to cluster the estimates in the middle of the array, whereas MT and GC are able to produce more expressive estimates that appear closer to the actual neuron locations. Since we have access to the ground truth data, we are able to quantitatively benchmark the results beyond visual inspection. The metrics we benchmark for each algorithm are accuracy (percentage of estimated locations within a specified radius of the true location), localization error (Euclidean distance between the estimated location and true location), and runtime. In the experimental dataset, since the ground truth location is known with lower resolution (there is a margin of error in the instrument calibration used to obtain the data, and we only know location of nearest electrode), we allow a larger radius for calculating "accurate" predictions.

In the simulated dataset (Fig 2C), GC and MT perform better than COM. This is consistent with our visual analysis, as COM's tendency to cluster estimates toward the center of the array results in lower-quality estimates. This limitation is more pronounced in our selected probe geometry (Neuropixel 384-channel probe), where the electrode array is roughly 4mm in length and 60μm in width, and so approximates a "linear" one-dimensional probe. In a one-dimensional probe, the convex hull of the array narrows to a line and COM is only able to estimate locations along this line, whereas the more physically realistic qualities of GC and MT leverage the relative signal strength along the line to infer locations in the dimensions orthogonal to the linear probe. In the experimental dataset (Fig 2D), COM performs materially better than MT and marginally better than GC. Our belief as to why this occurs is that the experimental dataset is inherently noisier, and COM is more robust to noise than the other algorithms (as we will see in electrode degradation). Lastly, we note COM is fastest in the simulated dataset, and MT is substantially slower as it must solve an optimization problem. Speed is important in spike localization (but less so for templates) given the higher volume of spikes and the need to localize quickly for real-time spike sorting, potentially on embedded devices with limited computing power (v. templates which can be analyzed post-sorting) [7,30,31].

To assess the statistical significance of performance differences across algorithms, we conducted one-way ANOVA tests for each metric followed by Tukey's HSD post-hoc comparisons. All metrics exhibited significant differences, though not all pairwise comparisons were significant. For template and spike accuracy, both GC and MT significantly outperformed COM ($p < 0.01$), while differences between MT and GC were not statistically significant. For template and spike

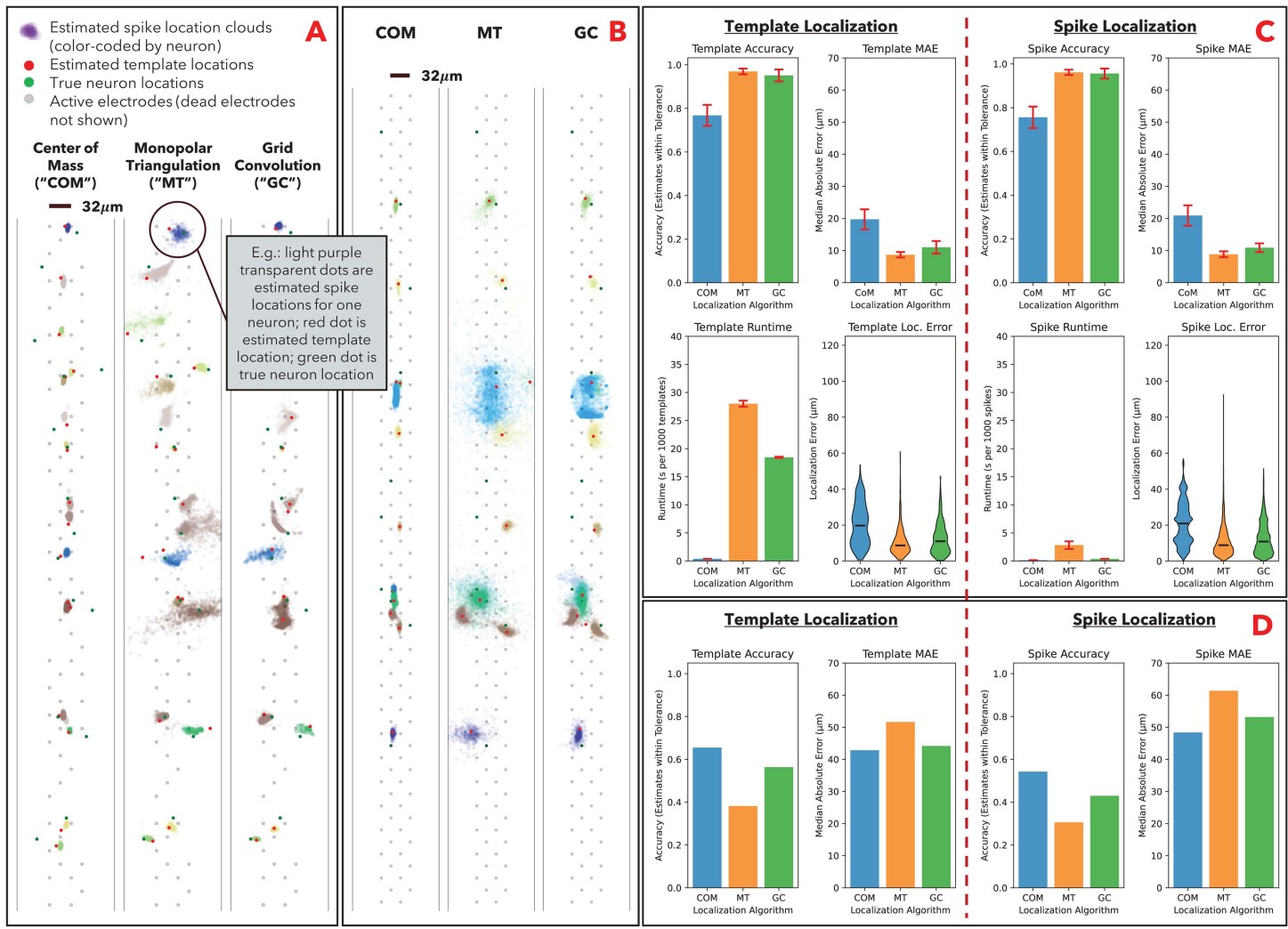

**Fig 2. SSL algorithm performance with no electrode degradation. A**: Visualization of template and spike localization estimates on simulated dataset (MEArec). The estimated spike locations are represented by translucent dot clouds, color-coded by their corresponding neuron. The estimated template location and true neuron locations are represented by solid red and green dots, respectively, and are not color-coded by neuron. The electrode array (shown excluding the "dead" electrodes to illustrate electrode degradation) is denoted by light gray dots. **B**: Visualization of template and spike localization estimates on experimental dataset (SPE-1). **C**: Performance metrics on simulated dataset, for both templates (left) and spikes (right). Performance is assessed as (i) percentage of estimates within 30μm of true locations (accuracy), (ii) speed of algorithm on all templates/spikes (runtime), and **(iii)** Euclidean distance between estimates and true location (median localization error and violin plot of individual localization errors). Bars represent mean metric across multiple simulations; error bars represent standard deviation of metric. Statistical significance was assessed using one-way ANOVA followed by Tukey's HSD post-hoc tests; all metrics exhibited significant differences (ANOVA p < 0.05). **D**. Performance metrics on experimental dataset, for both templates (left) and spikes (right). Performance is assessed using accuracy and runtime; since we have lower resolution of true locations in experimental data, we use a higher tolerance for accuracy (deemed correct if estimate within 50μm of estimated true location). We additionally do not include runtime, as the experimental neurons were recorded across different recording sessions (we concatenate onto one probe in Fig 2B for presentation purposes), which does not allow for localization algorithm to amortize certain overhead costs across neurons. Bars represent metric across all experimental data.

runtime, MT was significantly slower than both COM and GC (p < 0.01), confirming its computational cost. For template and spike median localization error, GC and MT again significantly outperformed COM (p < 0.01), while differences between MT and GC were not statistically significant.

## Benchmarking results against degradation

We next benchmark the localization algorithms in the case of signal loss; i.e., how robust the localization algorithms are to electrode degradation. In long-term neural recordings, individual electrodes will "fail" before the broader array, and over time the array will lose signal at an increasing proportion of its electrodes. In both the simulated ground truth and experimental ground truth datasets, we replace the signals of an increasing proportion of electrodes with Gaussian noise (the "dead electrodes").

The deterioration of localization estimates as a result of electrode degradation is visualized for the simulated dataset (Fig 3C) and experimental dataset (Fig 4C). We observe that COM is the most conservative method, in that signal loss

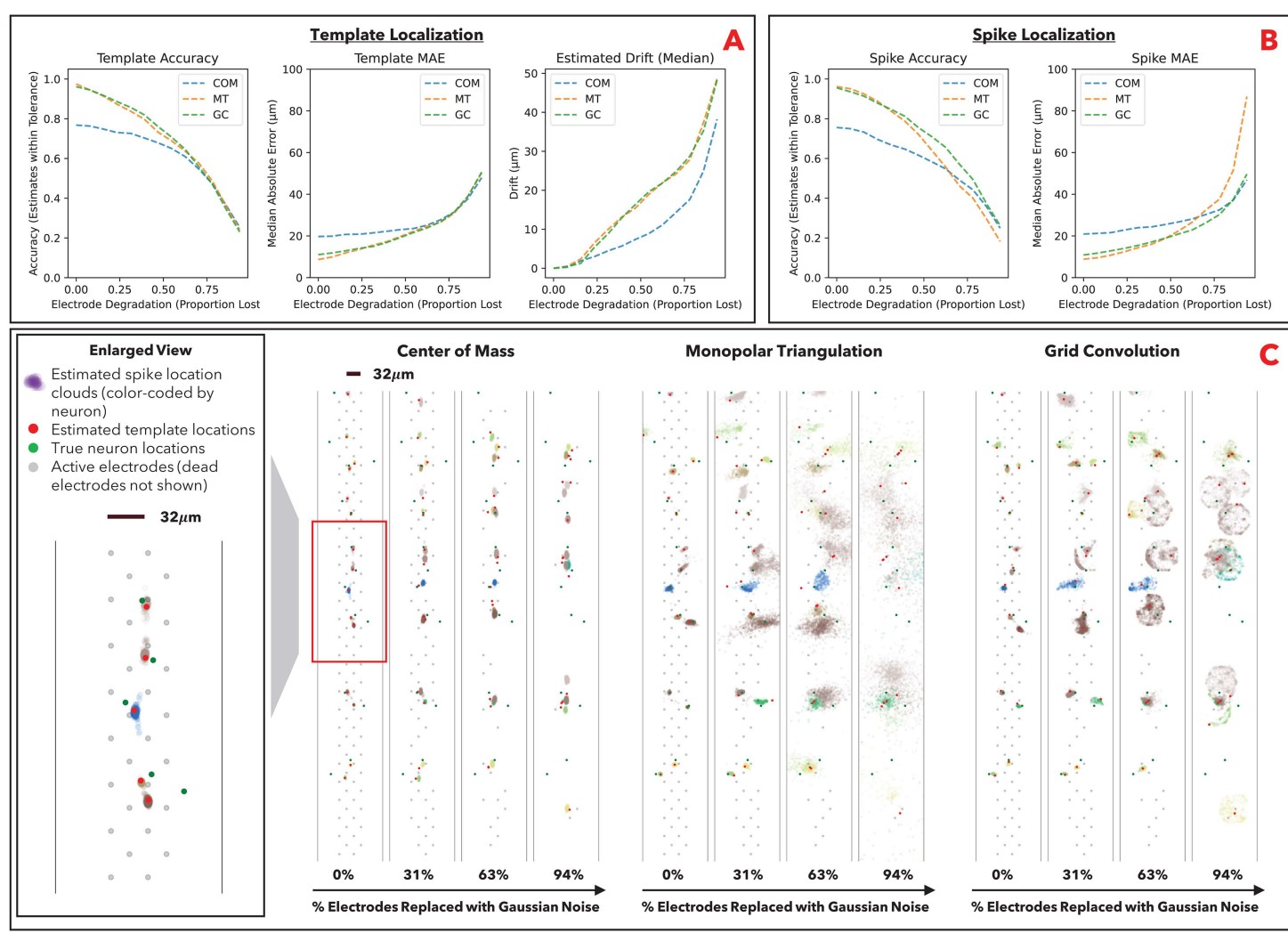

**Fig 3. SSL algorithm robustness on simulated data (MEArec). A**: Template localization performance metrics at varying levels of electrode degradation. Performance is assessed as (i) percentage of estimated locations within 30µm of true locations (accuracy), (ii) median Euclidean distance between estimated and true locations (localization error), and (iii) estimated drift from original location. Estimated drift is the Euclidean distance between the estimated template location without degradation v. the estimated template location with degradation; i.e., a measure of how unstable the localization algorithm is against degradation. **B**: Spike localization performance metrics at varying levels of electrode degradation. Performance is assessed via accuracy and localization error. **C**: Visualization of template and spike localization estimates at varying levels of electrode degradation. Left panel is zoomed-in view of visualization outlined in red.

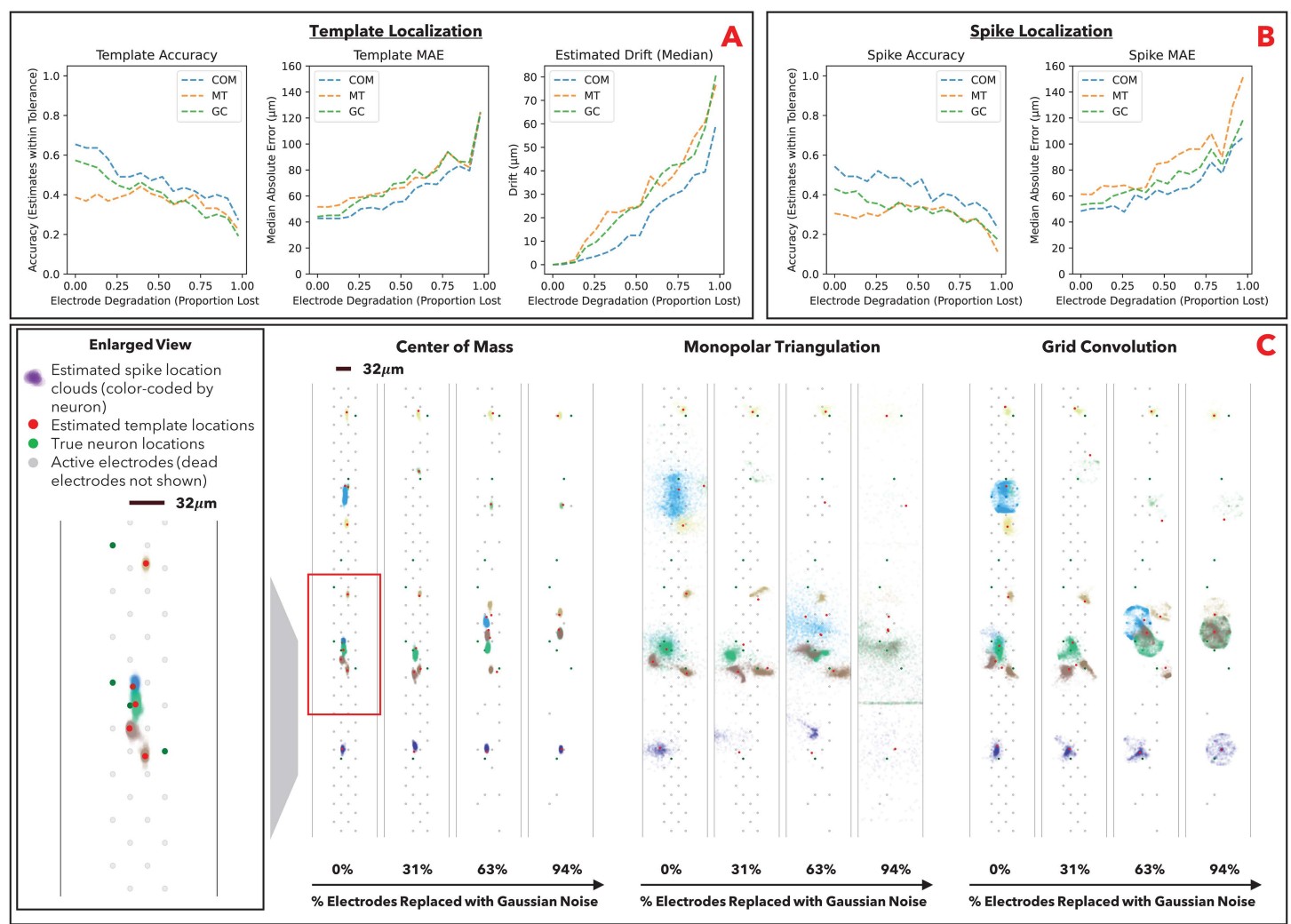

**Fig 4. SSL algorithm robustness on experimental data (SPE-1). A**: Template localization performance metrics at varying levels of electrode degradation (measured as the percentage of electrodes replaced with Gaussian noise). Performance is assessed as (i) percentage of estimated locations within 50μm of true locations (accuracy), (ii) median Euclidean distance between estimated and true locations (localization error), and (iii) estimated drift from original location. Since we have lower resolution actual location in experimental data v. simulated data, we use a higher tolerance for accuracy v. simulated data (50μm v. 30μm). **B**: Spike localization performance metrics at varying levels of electrode degradation. Performance is assessed via accuracy and localization error. **C**: Visualization of template and spike localization estimates at varying levels of electrode degradation. Left panel is zoomed-in view of visualization outlined in red.

narrows the convex hull of the remaining electrodes, and COM estimates increasingly congregate towards the center of the array. GC is also conservative in that the estimates congregate in the grid around the remaining electrodes; this follows from the algorithm relying on the convolution between actual and theoretical signals at each electrode, and the convolution will be less meaningful around dead electrodes (i.e., will devolve to trivial convolution with Gaussian noise); intuitively, the theoretical and actual signals will overwhelmingly match around live electrodes v. Gaussian noise. MT is the least conservative; the algorithm solves the optimization problem with progressively less information, resulting in overfitting and a larger number of highly erroneous estimates (particularly for individual spikes). Our benchmarking on the simulated ground truth dataset (Fig 3A and 3B) and experimental ground truth dataset (Fig 4A and 4B) similarly show

that COM and GC are more robust against electrode degradation than MT. Our analysis additionally includes a measure of estimated drift from the original location (Fig 3A and 4A). Estimated drift is the Euclidean distance between the estimated template location without degradation and the estimated template location with degradation; i.e., a measure of how unstable the localization estimates are against degradation. Stability is important in long-term recordings where we desire to consistently estimate neuron locations over time as electrodes degrade (erroneous drift can result in the same neuron being identified as different neurons over long-term recordings).

We note that MT and GC have a meaningful reliance on parameters regulating algorithm performance, including parameters regulating which channel signals are evaluated and the candidate window for solutions. While our results are based on the default values of these parameters in the SpikeInterface implementation, we have additionally performed a grid search on how different parameter sets may affect our study conclusions (Fig 5). We note that the study conclusions are robust to parameter choice. Across parameter choices in the simulated dataset, we continue to see that MT and GC perform better in the lower degradation conditions, and GC and COM perform better in higher degradation conditions. In the experimental dataset, we continue to see that COM performs marginally better than GC, and substantially better than MT across degradation conditions. We additionally observe that in each grid, the "upper left" represents higher regularization and the "lower right" represents lower regularization. For MT, the top left parameter set considers a smaller local radius of electrode signals (i.e., considers fewer, stronger signals) and limits the solutions to a smaller solution set. For GC, the top left parameter set also considers a smaller local radius of electrode signals and uses a smaller percentage of convolution results (i.e., fewer, stronger signals) in the weighted average. Under the lower degradation conditions in the simulated dataset, the less regularized parameter sets perform better than the more regularized parameter sets ("lower right" has lower errors than "upper left"); as electrode degradation increases, this observation "flips" and the more regularized parameters generally perform better than the less regularized parameters. This trend is also evident in the experimental dataset, though not as clearly indicated as the experimental dataset is inherently noisier (towards higher degradation conditions, we also see "lower right" performing worse in experimental dataset). This is consistent with our findings on how COM and GC perform better than MT under noisier conditions; when degradation and noise increase, the increased regularization is beneficial to accurate localization.

## Discussion

This study provides a first-of-its-kind ground truth benchmarking of three widely used spike source localization algorithms: center of mass, monopolar triangulation, and grid convolution. As the field of brain-machine interfaces experiences rapid growth and newer localization algorithms have been proposed with superior theoretical performance, it is important to ground these assessments in biological reality. Our study provides results for three algorithms, but also a ground-truth framework for benchmarking future localization algorithms.

### Key results & implications

The results of our benchmarking indicate significant differences in the performance of the three localization algorithms. In the absence of electrode degradation, GC and MT outperformed COM in the simulated ground truth dataset. This aligns with our expectations, as both GC and MT incorporate more physically realistic models compared to COM [32,33]. However, we see MT performance suffer in the experimental ground truth dataset, highlighting the importance of real-world biological conditions in assessing a localization algorithm. When accounting for the electrode degradation that occurs in long-term recording conditions, our findings demonstrate that COM and GC exhibit superior resilience compared to MT. As electrode failure rates increased, MT produced overfitting and larger localization errors, particularly for individual spike events. In contrast, COM's simplicity and GC's grid-based guardrails provided more stable estimates, making them more suitable for long-term recordings where electrode degradation is inevitable.

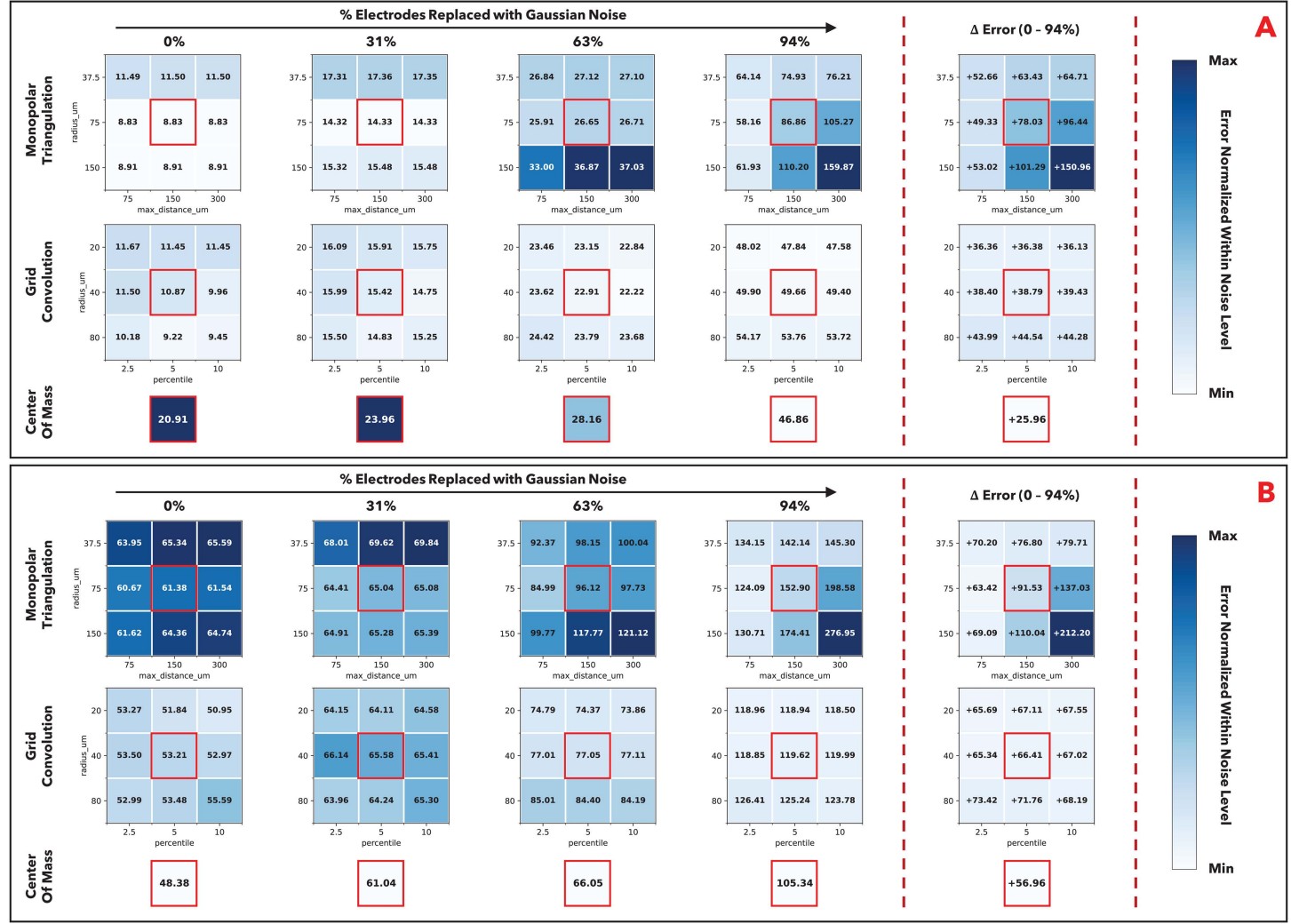

**Fig 5. Impact of localization parameters on median spike localization error.** For MT, we examine the parameters "radius_um" (denoting the radius around the peak channel where signal is considered in optimization) and "max_distance_um" (denoting the max distance from the COM initial estimate considered as a solution to optimization). For GC, we examine the parameters "radius_um" (similarly denoting radius around the peak channel where signal is considered in optimization) and "percentile" (denoting percentage of the best scalar products used to estimate location). For COM, the parameters have less meaningful impact on algorithm performance and so we present the median spike localization error under default parameters. Median error using the default parameters are marked in red boxes, and we examine parameters an order of two lower and higher than the default parameters. In the first four columns, we show the progression of the median localization error as electrode degradation increases, and the colors denoting magnitude of error are normalized within the individual electrode degradation level but across the three localization methods (i.e., normalized within the "column"). We additionally show the change in localization error from 0% to 94% degradation for each method and parameter set in the fifth column, to indicate how much electrode degradation impacts performance. **A**: Results on the simulated dataset (MEArec). **B**: Results on the experimental dataset (SPE-1).

The superior robustness of COM and GC suggest better utility in long-term neural recordings, where resilience to electrode degradation and consistent spike localization are crucial for accurate longitudinal analysis. Localization results using COM and GC are likely to be more informative in spike sorting (as well as faster to calculate), and should better discriminate consistent neuron populations over time. In addition, the more reliable performance results are a more appropriate metric for assessing probe drift.

## Future direction

We hope that the code for our benchmarking methodology, which has been publicly released and references publicly available datasets, will be informative for the continued assessment of improvements in spike source localization algorithms.

Moreover, we believe that spike source localization is a highly nascent field, where the use-cases are core to the advancement of brain-machine interfaces, but current algorithms are either based on simple heuristics or not sufficiently robust to biological noise. We note that at its core, the spike source localization problem is not new; it is a "triangulation" problem, which has been widely explored in fields such as geospatial localization techniques in telecommunications and geology. Methods from geospatial applications, such as fuzzy logic and time-of-difference arrival techniques, have demonstrated superior performance in determining locations of wireless signal sources in real-world settings [34–39]. By integrating principles from these fields, we believe there is significant opportunity to substantially improve the performance of spike source localization algorithms in brain-machine interfaces.

## Acknowledgments

All authors would like to thank the Liu Group for helpful discussions. Computations were performed using resources provided by Harvard University.

## Author contributions

**Conceptualization:** Hao Zhao, Xinhe Zhang, Arnau Marin-Llobet, Jia Liu.

**Data curation:** Hao Zhao.

**Formal analysis:** Hao Zhao.

**Investigation:** Hao Zhao.

**Methodology:** Hao Zhao, Xinhe Zhang, Arnau Marin-Llobet.

**Project administration:** Hao Zhao, Xinhe Zhang.

**Resources:** Hao Zhao, Arnau Marin-Llobet, Jia Liu.

**Software:** Hao Zhao.

**Supervision:** Hao Zhao, Xinhe Zhang, Jia Liu.

**Validation:** Hao Zhao.

**Visualization:** Hao Zhao.

**Writing – original draft:** Hao Zhao.

**Writing – review & editing:** Hao Zhao, Xinhe Zhang, Arnau Marin-Llobet, Xinyi Lin.

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
