## [Decision Letter · Decision Letter 0]

15 Oct 2025

PCOMPBIOL-D-25-01653

Benchmarking spike source localization algorithms in high density probes

PLOS Computational Biology

Dear Dr. Zhang,

Thank you for submitting your manuscript to PLOS Computational Biology. After careful consideration, we feel that it has merit but does not fully meet PLOS Computational Biology's publication criteria as it currently stands. Therefore, we invite you to submit a revised version of the manuscript that addresses the points raised during the review process.

Specifically, in addition to requesting a better description of the methods, the reviewers point out that this work would be stronger, and closer to the experimental data, if also a Neuropixel probe was simulated.

Please submit your revised manuscript within 60 days Dec 15 2025 11:59PM. If you will need more time than this to complete your revisions, please reply to this message or contact the journal office at ploscompbiol@plos.org. Please include the following items when submitting your revised manuscript:

We look forward to receiving your revised manuscript.

Kind regards,

Matthias Helge Hennig, Ph.D.

Academic Editor

PLOS Computational Biology

Daniele Marinazzo

Section Editor

PLOS Computational Biology

**Journal Requirements:**

At this stage, the following Authors/Authors require contributions: Xinyi Lin, Xinhe Zhang, Jia Liu, Arnau Marin-Llobet, and Hao Zhao. Please ensure that the full contributions of each author are acknowledged in the "Add/Edit/Remove Authors" section of our submission form.

4) Please provide a detailed Financial Disclosure statement. This is published with the article. It must therefore be completed in full sentences and contain the exact wording you wish to be published.

1) Please clarify all sources of financial support for your study. List the grants, grant numbers, and organizations that funded your study, including funding received from your institution. Please note that suppliers of material support, including research materials, should be recognized in the Acknowledgements section rather than in the Financial Disclosure

2) State the initials, alongside each funding source, of each author to receive each grant. For example: "This work was supported by the National Institutes of Health (####### to AM; ###### to CJ) and the National Science Foundation (###### to AM)."

3) State what role the funders took in the study. If the funders had no role in your study, please state: "The funders had no role in study design, data collection and analysis, decision to publish, or preparation of the manuscript."

4) If any authors received a salary from any of your funders, please state which authors and which funders..

5) Please ensure that the funders and grant numbers match between the Financial Disclosure field and the Funding Information tab in your submission form. Note that the funders must be provided in the same order in both places as well.

6) Please send a completed 'Competing Interests' statement, including any COIs declared by your co-authors. If you have no competing interests to declare, please state "The authors have declared that no competing interests exist". Otherwise please declare all competing interests beginning with the statement "I have read the journal's policy and the authors of this manuscript have the following competing interests"

**Reviewers' comments:**

Reviewer's Responses to Questions

**Comments to the Authors:**

Reviewer #1: Review uploaded as an attachment.

Reviewer #2: The theme is interesting and important, and the manuscript is decent and technically sound. It is, however, not very thorough, and has room for improvement. Below are a few comments and suggestions.

1) The Methods section could be expanded. For example, what database of cell models are used? Are the cell models a mix of pyramidal cells and interneurons?

2) It should be relatively simple to emulate the Neuropixel probe in the simulations, instead of using a generic square probe. This would make simulated and experimental verification easier to compare. Do the authors expect the MT to perform worse than GC and COM in this case, or only in the actual experimental data (as stated in Line 191)? One could then also perhaps test the effect of not knowing the exact location of the soma (only the nearest electrode) in the experimental data, since this might also affect conclusions from the localization.

3) Line 58: "Finally, uncorrelated Gaussian noise with 1 mV standard deviation was added to the traces." This number seems extremely high, and is probably wrong. I would have expected the noise level to more like 10 µV?

4) Line 55: "30 nm apart" Is the unit here wrong perhaps?

5) Line 126: "(Neuropixel 384-channel), which is 5 mm in length and 40 µm in width" Are these numbers correct? They seem different than previous numbers I have seen.

6) These two papers might be relevant for the discussion, since they also work on spike localization in high density probes:

Delgado Ruz & Schultz (2014). Localising and classifying neurons from high density MEA recordings. Journal of Neuroscience Methods, 233, 115–128.

Buccino et al (2018). Combining biophysical modeling and deep learning for multi-electrode array neuron localization and classification. Journal of Neurophysiology, 120, 1212–1232.

**Have the authors made all data and (if applicable) computational code underlying the findings in their manuscript fully available?**

Reviewer #1: Yes

Reviewer #2: Yes

PLOS authors have the option to publish the peer review history of their article (what does this mean? ). If published, this will include your full peer review and any attached files.

**Do you want your identity to be public for this peer review?** For information about this choice, including consent withdrawal, please see our Privacy Policy .

Reviewer #1: **Yes:** Olivier Winter

Reviewer #2: No

**Figure resubmission:**
---

## [Decision Letter · Decision Letter 1]

9 Jan 2026

PCOMPBIOL-D-25-01653R1

Benchmarking spike source localization algorithms in high density probes

PLOS Computational Biology

Dear Dr. Zhang,

Thank you for re-submitting your manuscript to PLOS Computational Biology. After careful consideration, we feel that it has merit, but reviewer 1 has raised two valid issues (data sharing and parameter sweeps) that should be addressed before we can accept the paper. Therefore, we invite you to submit a revised version of the manuscript that addresses these points.

We look forward to receiving your revised manuscript.

Kind regards,

Matthias Helge Hennig, Ph.D.

Academic Editor

PLOS Computational Biology

Daniele Marinazzo

Section Editor

PLOS Computational Biology

**Reviewers' comments:**

Reviewer's Responses to Questions

**Comments to the Authors:**

Reviewer #1: The authors have addressed the couple of large revisions requested. Namely, (1) on the geometry of the synthetic dataset electrodes raised by both reviewers, they regenerated and re-ran the algorithm on a synthetic dataset matching the electrode configurations of the experimental dataset. (2) They addressed the comment about the noise levels on synthetic data and set levels more in line with experimental data (note that to be thorough, the noise should not be white or gaussian, but have a non-flat PSD like pink noise, yet this is perfectly publishable in this state and I should have raised it on the original review).

They clarified the pre-processing by adding to the methods and removed claims about run-time.

There remains only minor revisions and a suggestion. For the minor revisions:

- On the data sharing, I think my comment has been misunderstood. As of now, someone wanting to reproduce those results has to download terabytes of data and re-run a non-trivial pre-processing pipeline. I was suggesting to provide spike trains and average waveforms, with the corresponding localisation results. For 11 cells this would hardly be more than a few tens of megabytes, and not 4 terabytes as the authors have commented. Yet it would make the dataset much more accessible to anyone wanting to play with localisation algorithms.

Going further, providing 256 - 512 individual spike waveforms per cell would barely reach hundred megabytes, and again prove a much more approachable dataset to anyone wanting a quick look in the matter instead of wading through the non-trivial pre-processing.

- l. 94 I wouldn't qualify a 300-3000 Hz bandpass filter standard, as the highcut frequency is unusual: it is standard in kilosort to use a 3d order 300 Hz highpass applied forward and backward (making it a sixth order zero-phase filter). Remove "standard" and state the filter order and if it is causal or zero-phase.

- l. 60 Aggregation: which one ? It could be a count, the min, max, the average or median. I suspect it is the median or average but worth being more precise !

For the suggestion, I still think it would be nice to have some grid search of the localisation parameters. Here the authors are relying on spikeinterface default parameters, and dismissed the recommendation of exploring meta-parameters as the study being only within the scope of this standard use. Yet I would enjoin the authors to be a bit more careful about assuming adequacy of those default parameters, and to be frank, I would also enjoin them to be more curious. For example, what if your results exhibit a large sensitivity to those parameters ? Not only that would diminish some of this study conclusions, but that would also mean optimizing the parameter search may be an interesting avenue for improving the localisation method.

Looking forward to seeing this published !

Reviewer #2: The authors have responded well to all of my concerns, and I have no further comments. I recommend that the manuscript is accepted.

**Have the authors made all data and (if applicable) computational code underlying the findings in their manuscript fully available?**

Reviewer #1: Yes

Reviewer #2: Yes

PLOS authors have the option to publish the peer review history of their article (what does this mean? ). If published, this will include your full peer review and any attached files.

**Do you want your identity to be public for this peer review?** For information about this choice, including consent withdrawal, please see our Privacy Policy .

Reviewer #1: **Yes:** Olivier Winter

Reviewer #2: No

**Figure resubmission:**
---

## [Editor Report · Decision Letter 2]

24 Feb 2026

Dear Zhang,

We are pleased to inform you that your manuscript 'Benchmarking spike source localization algorithms in high density probes' has been provisionally accepted for publication in PLOS Computational Biology.

Best regards,

Matthias Helge Hennig, Ph.D.

Academic Editor

PLOS Computational Biology

Daniele Marinazzo

Section Editor

PLOS Computational Biology

---

## [Editor Report · Acceptance letter]

PCOMPBIOL-D-25-01653R2

Benchmarking spike source localization algorithms in high density probes

Dear Dr Zhang,

I am pleased to inform you that your manuscript has been formally accepted for publication in PLOS Computational Biology. Your manuscript is now with our production department and you will be notified of the publication date in due course.

With kind regards,

Anita Estes
